# Controlling technical variation amongst 6693 patient microarrays of the randomized MINDACT trial

Laurent Jacob [1], Anke Witteveen[2], Inès Beumer[2], Leonie Delahaye[2], Diederik Wehkamp[2], Jeroen van den Akker[2], Mireille Snel[2], Bob Chan [2], Arno Floore[2], Niels Bakx[2], Guido Brink[2], Coralie Poncet[3], Jan Bogaerts[3], Mauro Delorenzi [4,5], Martine Piccart[6], Emiel Rutgers[7], Fatima Cardoso [8], Terence Speed [9], Laura van 't Veer [2,10✉] & Annuska Glas [2✉]

Gene expression data obtained in large studies hold great promises for discovering disease signatures or subtypes through data analysis. It is also prone to technical variation, whose removal is essential to avoid spurious discoveries. Because this variation is not always known and can be confounded with biological signals, its removal is a challenging task. Here we provide a step-wise procedure and comprehensive analysis of the MINDACT microarray dataset. The MINDACT trial enrolled 6693 breast cancer patients and prospectively validated the gene expression signature MammaPrint for outcome prediction. The study also yielded a full-transcriptome microarray for each tumor. We show for the first time in such a large dataset how technical variation can be removed while retaining expected biological signals. Because of its unprecedented size, we hope the resulting adjusted dataset will be an invaluable tool to discover or test gene expression signatures and to advance our understanding of breast cancer.

[1] Université de Lyon, Université Lyon 1, CNRS, Laboratoire de Biométrie et Biologie Évolutive UMR 5558, Villeurbanne, France. [2] Agendia NV/Agendia Inc, Amsterdam, The Netherlands. [3] EORTC, Brussels, Belgium. [4] University Lausanne, Lausanne, Switzerland. [5] SIB Swiss Institute of Bioinformatics, Lausanne, Switzerland. [6] Institute Jules Bordet, Brussels, Belgium. [7] Netherlands Cancer Institute, Amsterdam, The Netherlands. [8] Champalimaud, Lisbon, Portugal. [9] Walter and Eliza Hall Institute of Medical Research, Melbourne, VIC, Australia. [10] Helen Diller Family Comprehensive Cancer Center, University California San Francisco, San Francisco, CA, USA. ✉email: Laura.Vantveer@ucsf.edu; annuska.glas@agendia.com

Gene (mRNA) expression data of tumor tissue biospecimens that are collected over an extended period of time during for instance large clinical trials, are susceptible to variation due to factors such as adjustments in protocols, equipment, or batches of reagents. Transparency and managing the potential influence of these technical factors on expression profiles are critical to avoid spurious conclusions[1].

The largest trial that collected full-transcriptome expression data is the MINDACT trial[2,3], an international, multi-center, prospective, phase III randomized trial whose main objective was to prospectively validate the 70-gene MammaPrint signature as a prognostic tool for the clinical evaluation of patients to help avoid chemotherapy[4]. Trial results showed that the 70-gene expression signature enables a reduction of 46% in the use of chemotherapy for clinically high-risk patients[4], a finding that was adopted in various international breast cancer guideline recommendations[5–9].

During the 5-year enrollment phase of MINDACT, the measurement of the 70-gene expression signature was performed under US Food and Drug Administration Quality System Regulation (FDA/QSR)-compliant protocols, which included technical adjustments and improvements (Fig. 1a). As such, technical variations were controlled to provide a robust MammaPrint test result based on the built-in normalization genes specific for MammaPrint and quality control methods[4,10–12]. In addition, the risk classification using the 70 genes is binary and is a result of the combination of the expression of the 70 genes. Here, we are studying all other genes and the procedures to monitor and guarantee the risk classification of MammaPrint, however, did not monitor or guarantee that the full-transcriptome data was unaffected by technical variation.

Initial steps in standard microarray procedures and as employed for this dataset based on a two-color method, convert raw feature extracted image data to biological significant data on a per gene-probe measurement at full-transcriptome level. This includes a multi-step process involving quality control per probe, as well as normalization steps that correct for differences in sample labeling efficiency or the amount of RNA used. These

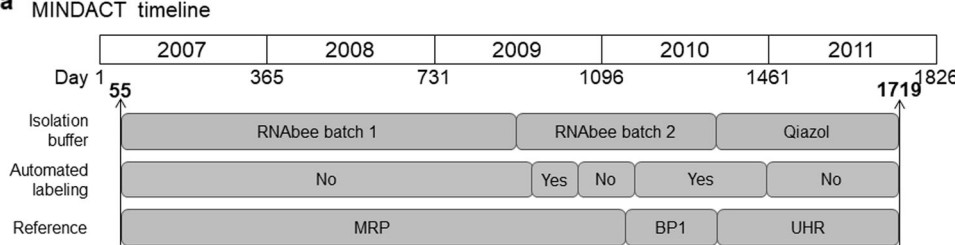

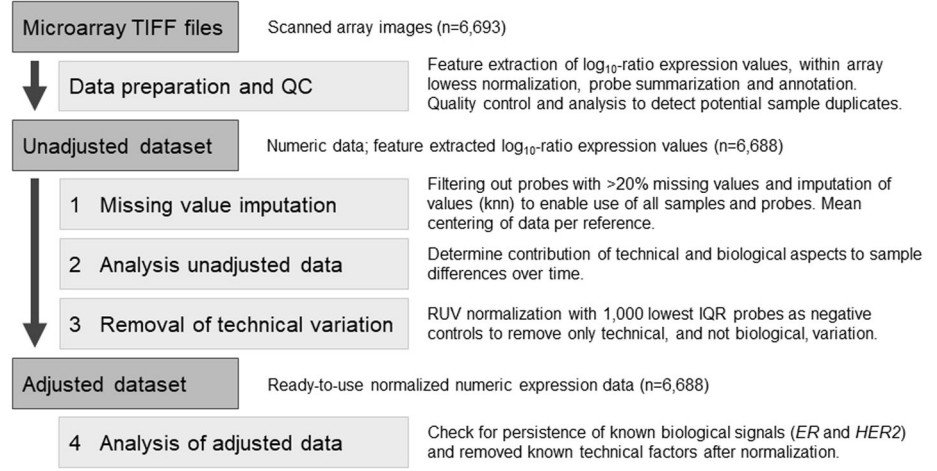

**Fig. 1 MINDACT timeline and full-transcriptome dataset analysis procedure. a** contains a timeline overview of the MINDACT enrollment and gene expression microarray technical adjustments and improvements. The MINDACT trial enrolled 6693 patients from February 2007 until September 2011 and samples were processed at Agendia's central laboratory. During this time, some standard operating procedures underwent FDA/QSR-compliant technical adjustments and improvements. Evaluation and calibration were performed to ensure accuracy of the MammaPrint test. The technical changes listed here were the main factors included in the assessment of variance on full-transcriptome data of the MINDACT samples, further detail and other factors evaluated are described in detail in the "Methods" section and Supplementary Table 1. The x-axis on top represents the years and the number of days between January 1st of 2007 (reference date) and the isolation date of the sample. This reference date was chosen to make the plots easier to read. Indicated are the first day (55) and the last day (1719) of patient sample isolation. Isolation buffer: commercial solution to extract RNA from (tumor) tissue, RNAbee (Teltest, TX, USA), and Qiazol (Qiagen, Germany), respectively. Reference is a standard RNA sample hybridized to the microarray together with each tumor RNA sample. Three different ones were used: MRP, Mamma Reference Pool, and BP1, Breast Pool 1, both made by Agendia from RNA isolated from a custom series of breast cancer samples and UHR, Universal Human Reference, a commercial RNA from human tumor reference samples (Agilent, USA). **b** shows the step-wise procedure of the MINDACT analysis pipeline for evaluating and managing technical variation in the gene expression levels representing the full-transcriptome of the MINDACT patients. For more detail see "Methods" and Supplementary Table 1. ER estrogen receptor, HER2 erb-b2 receptor tyrosine kinase 2, IQR interquartile range, knn k-nearest neighbor, n number, RUV remove unwanted variation, TIFF tagged image file format, QC quality control.

processes are performed per array ("within-array normalization"), however, they do not correct for variation caused by technical changes over time (such as changes in reagent batches, or equipment). Statistical methods[13–16] could be used that handle the unwanted technical variation when studying a factor of interest (e.g., therapy response prediction), however, one should be aware that analyses (e.g., investigating new subtypes) can be misled by unwanted variation, and prior adjustment of the data is warranted. Deciding what should be adjusted for is, however, a challenge, as yet unknown subtypes may be confounded with some technical factors and adjusting for the technical factors may discard the unknown subtypes. Identifying the technical variation is a non-trivial task, as part of it may be caused by undetected factors[17–26]. Accordingly, the objective of this paper is twofold. First, we aim to describe the unwanted variation affecting the expression data of this large MINDACT dataset. Principal component analyses suggested that time-varying technical factors affected the measured gene expression, but still showed the expected biological signals related to the main breast cancer subtypes. Secondly, our aim was to build an adjusted version of the expression dataset for future analysis purposes. We relied on an adaptation of the *random naive RUV* algorithm[27] to remove variation related to time-localized technical factors and mediate the variation caused by unobserved technical factors, without losing biological signals.

After adjustment, the top principal components are stable over time, and correlate with expected biological signals. The analysis pipeline is entirely reproducible as we provide the R scripts that were used to generate the tables and figures of this paper. The adjusted and unadjusted breast cancer datasets with unprecedented sample sizes are available through the trial sponsor, the MINDACT committee of European Organization for Research and Treatment of Cancer (EORTC), and ready-to-use for translational research, including discovery of new classifiers or testing of proposed signatures. Additionally, the step-wise analysis pipeline we describe provides guiding principles for the analysis and management of technical variation and big data processing of other large cohort studies spanning extended time periods of patient enrollment.

## Results

Tumor samples from breast cancer patients from 112 institutes located in nine countries who were enrolled in the MINDACT trial[4] (6693 patients) were processed at Agendia between February 2007 and September 2011, using customized full-transcriptome two-color Agilent microarrays that contained the MammaPrint probes (Fig. 1a and "Methods"). Technical adjustments and improvements made during the study, shown in Fig. 1a and in further detail described in the "Methods", were in compliance with FDA/QSR regulation for accurate MammaPrint readout. Here, we describe the management of variation observed in the full-transcriptome data and evaluate which factors could explain a substantial proportion. The effect of the technical factors (the unwanted technical variation) can be corrected, while the biological signals are preserved. The adjusted dataset we present is a prospectively collected gene expression dataset from a representative stage I and II breast cancer patient population that is normalized and available for further translational research.

The analysis pipeline that we used to manage the technical variation consists of data preparation of the raw data in prepossessing steps (QC, "within-array normalization"), and successive steps as depicted in Fig. 1b, which created the dataset adjusted for technical variation. The procedures of each step are described in the next paragraphs, as well as in the "Methods" section.

**Data preparation and quality control.** The basis of the gene expression measurements are TIFF files, scanned microarray images. Data preparation of the 6693 samples included standard feature extraction procedures, annotation and quality control (see "Methods"). Expression values of the dual channel patient plus reference sample microarray hybridizations are presented as $\log_{10}$-ratio values. During the MINDACT study three different RNA pools were used as reference ("Methods", data preparation). The full-transcriptome expression data of five patients was excluded based on poor quality, resulting in 6688 full-transcriptome study samples. In addition, between patient sample correlations were calculated to assess whether some samples may have been accidentally repeated (see "Methods" and Supplementary Note 1). Two patient samples with a high correlation to other patient samples were identified. Reprocessing of the samples showed that they were subjected to mix-ups preceding the hybridization step (the unadjusted and adjusted datasets of this study contain the corrected reprocessed versions of these two samples). Thirteen other patient samples with high between sample correlation to at least one other patient sample were also reprocessed for quality review. Their correlation remained high in the reprocessed data, and their distinct MammaPrint indices were consistent with their original versions, which confirms that they are not subject to a mix-up, and suggests that high correlations do not always indicate sample repeats (see Supplementary Note 1).

**Missing value imputation (step 1).** For the 6688 patient samples (40,793 probes per microarray), the number of missing values was determined (see Supplementary Fig. 1). Probes that were missing in >20% of the samples were omitted ($n = 5182$); more precisely as three different RNA references were used, missing values were determined per RNA reference group (see "Methods"), leaving 35,611 probes per patient for further analysis. The remaining missing values were imputed, and the data were zero-centered for each probe in each of the three reference groups to allow for combining the three sets of samples with different references for further analyses (see "Methods"). The total dataset after missing value imputation consists of 6688 samples with 35,611 data points.

**Analysis of the unadjusted expression data (step 2).** The next step was to assess the effect of known technical changes (listed in Supplementary Table 1) and to investigate expected biological variation (see "Methods"). Factors that strongly affect gene expression, such as different RNA isolation buffers used, may account for a substantial proportion of the total variance and are typically associated with the top principal components (PCs). In this dataset, the first PC (PC1) accounted for 12% of the total variance in the unadjusted dataset (Fig. 2a). The top four PCs (PC1–PC4) together accounted for 28% of the variance observed, while each following PC explained <3 additional percent of variance. Thus, we felt confident that the top four PCs summarize all major effects that could contribute to technical variance. We plotted the PCs against the isolation date of the samples in Fig. 3. Biological signals were expected to be more stationary over time, while technical signals could fluctuate, and be related to the time period when change was implemented.

The first PC (PC1) of the unadjusted data showed a sharp variation over time (Fig. 3a–c) and this suggested that it was mainly represented by technical rather than biological variation. As expected, most of this variation could be explained by a combination of technical factors related to RNA labeling, labeling yield, automated labeling, and the combination of which isolation buffer and reference was used. The two peaks in the distribution correspond to time periods when the labeling yield was stronger,

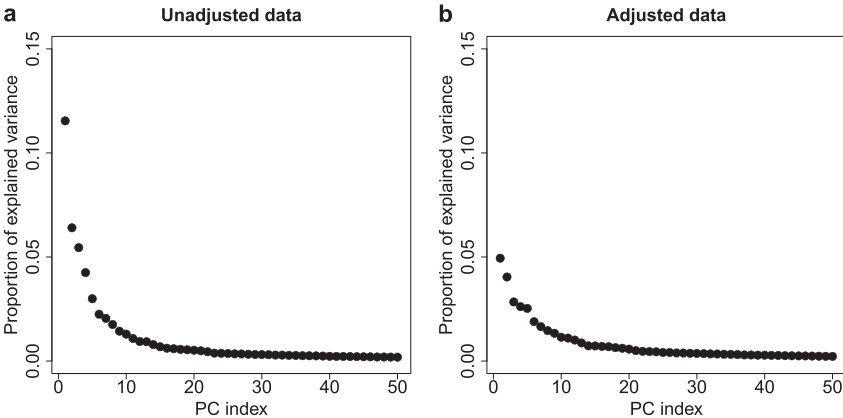

**Fig. 2 Proportion of variance explained by principal components before and after data adjustment.** Principal components (PCs) were computed across the $n = 6688$ patient samples over the 35,611 probes that remained after imputation of missing values (for further details see "Methods" section). Dots represent the variance explained by the respective PCs for each of the datasets (e.g., PC1 explains ~12% of the variance in the unadjusted dataset (**a**) and 5% of the variance in the adjusted dataset (**b**)). Data adjustment was aimed at removing the effect of technical factors from the gene expression while preserving biological signals. The figure shows that the proportion of variance explained by the top four PCs was reduced after the data adjustment (**a** vs. **b**). PC principal component.

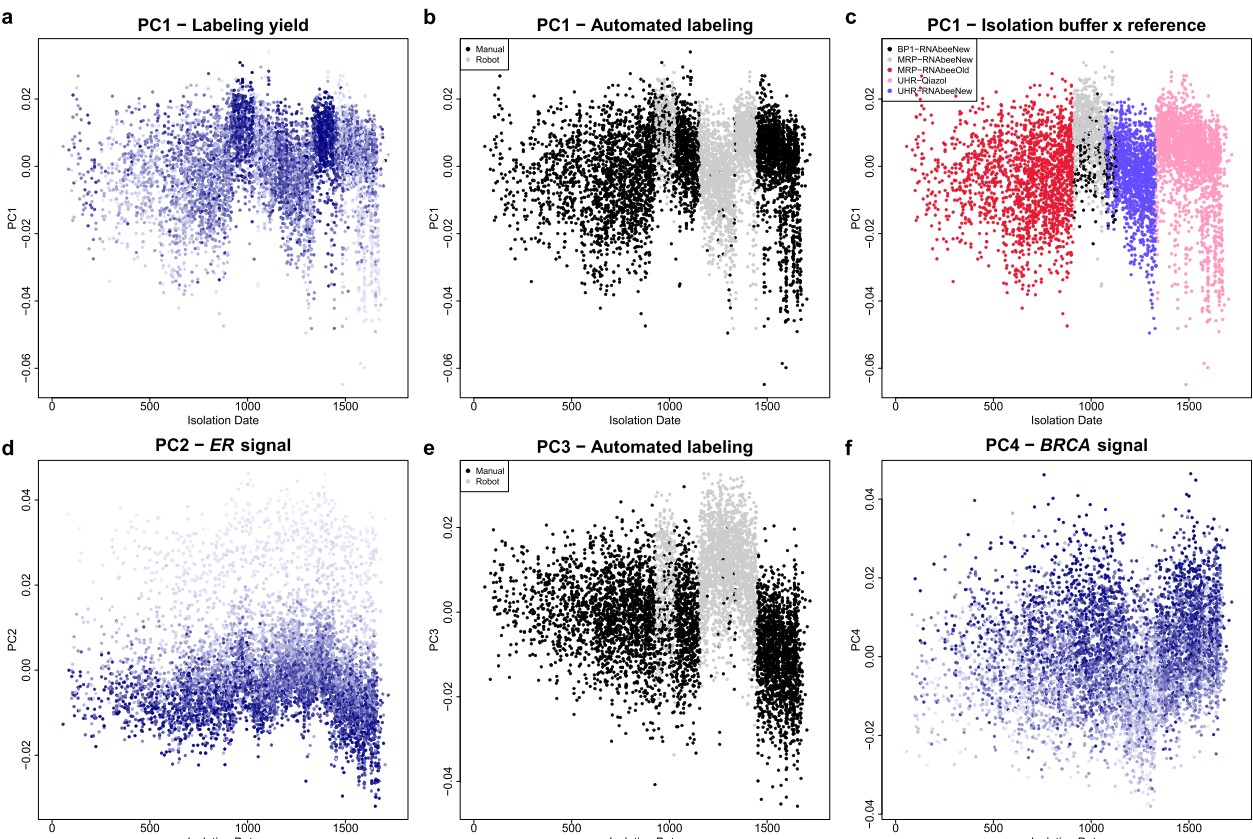

**Fig. 3 Variation over time of the top four principal components and their association with technical and biological factors in the unadjusted dataset.** Factors that strongly affect the observed gene expression account for a substantial proportion of the total variance and are typically associated with the top principal components (PCs). With the PC analysis we captured the largest proportion of variance across all arrays represented by all their probes in the unadjusted dataset. PC scores for the 6688 patient samples before data adjustment were plotted against time to visualize any variation of the technical or biological factors over time. Each dot represents one patient sample. The x-axis represents the number of days between January 1st of 2007 and the isolation date of the sample. The y-axis represents the projection of the samples onto one of the top four principal components. **a**–**c** show the projection of PC1 and are color-coded by labeling yield (blue intensity), the combination of the labeling buffer and the RNA reference used (see legend box), and the use of the robot for labeling or manual labeling (see legend box). PC2 projections are color-coded for *ER* signal (**d**; mRNA expression level from microarray), PC3 projections are color-coded for automated labeling (**e**), and PC4 for *BRCA* signal (**f**; mRNA expression level from microarray). For plots without color coding legends, the color intensity of the blue dots represents the value of the plotted signal, with high color intensities corresponding to high signals ("Methods"). *BRCA* breast cancer gene expression, *ER* estrogen receptor, PC principal component.

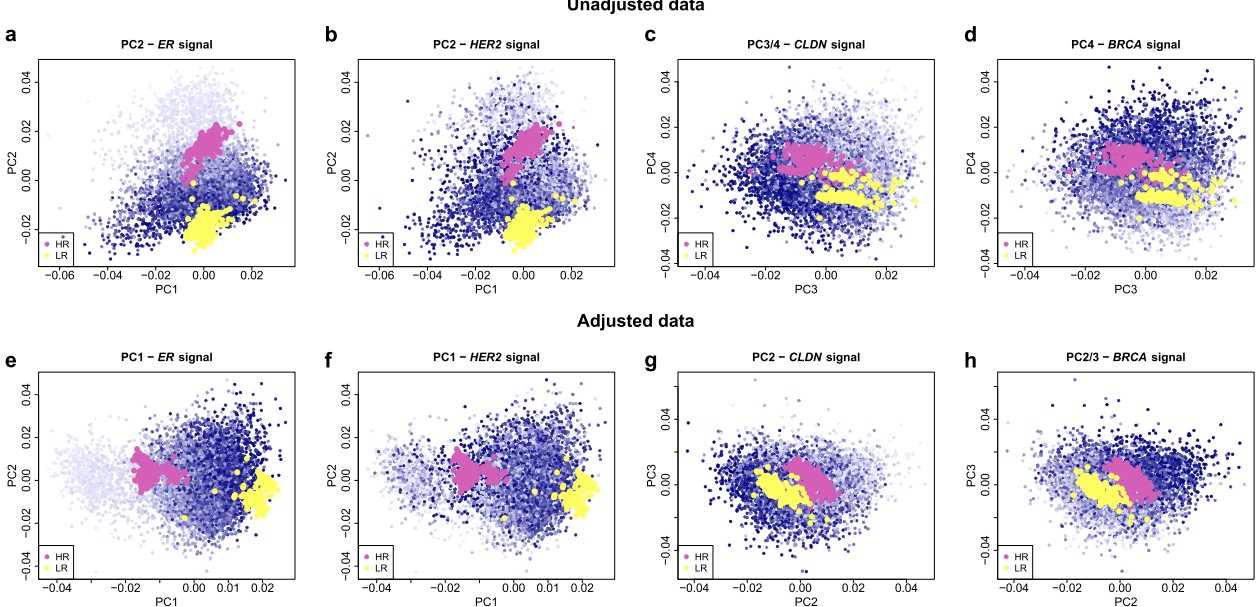

**Fig. 4 Principal component projections of the samples before and after adjustment and their association with biological signals.** A factor that strongly affects the observed gene expression accounts for a substantial proportion of the total variance across all arrays and is typically associated with the top principal components (PCs), as these are ordered by the proportion of variance they catch. We calculated the PCs separately in the unadjusted and the adjusted dataset. Because of the substantial data transformation performed by the adjustment procedure, the PC scores cannot be compared in a one to one fashion. The plots serve to show that the association between the biological signals and the PCs changes as a result of the data adjustment. In particular, due to the disappearance of technical variation, the association of the biological signals shift to more prominent PCs, as expected. Scatterplots of the 6,688 patient samples projected onto two-dimensional PC planes (PC1 vs PC2, PC2 vs PC3 or PC3 vs PC4) are shown before (**a**–**d**) and after (**e**–**h**) data adjustment, with color intensity, indicating the value of the plotted biological signal and high color intensities corresponding to high signals (see "Methods"). Individual plot titles indicate the PC that is most associated with the particular biological signal, as indicated by a gradient of blue color intensity along the PC. Biological variation in the data was not lost by the adjustment, as the gradient remained visible in the adjusted data and associated with a more prominent PC (**e**–**h**). This is also shown by the 562 control samples (see "Methods") that were not used to calculate the PCs but were added to the plots. The yellow dots represent three different MammaPrint low-risk controls, the purple dots five different high-risk controls. These eight controls also differ in the breast cancer subtype they belong to; they are separated well by PC2 before adjustment, and by PC1 after adjustment and they show a high consistency of the biological signals unaffected by the adjustment. *BRCA* breast cancer gene expression, *CLDN* claudin, *ER* estrogen receptor, *HER2* erb-b2 receptor tyrosine kinase 2, *PC* principal component.

which can be deduced from the more intense blue coloring of the dots representing the individual samples in Fig. 3a. The two peaks also correlated automated labeling with PC1 (Fig. 3b) and the combination of isolation buffer and reference RNA (Fig. 3c). Samples with previously defined MammaPrint scores that were used as technical and experimental controls within each batch run of MammaPrint were processed on the same array type (methods). Their projections to PC1 follow the same fluctuation patterns, thereby confirming their technical nature (see Supplementary Fig. 2). Moreover, Pearson correlations between PC1 and *BRCA* and *ER* expression were 0.43 and 0.2, respectively, and showed no clear time structure confirming that variation in PC1 was mainly carried by technical factors.

The PC2 of the expression data mainly corresponded to biological signals that are more regular over time than the PC1 projections (Fig. 3d). This can also be seen in Fig. 4, where the samples were plotted towards combinations of the top four PCs and color-coded for estimates of relevant biological signals such as *ER*, *HER2*, *CLDN*, and *BRCA* (see "Methods"). PC2 separated samples with low and high *ER* signals, thereby indicating the presence of the expected *ER*- and *ER* + subgroups[28] (Fig. 4a, light vs. dark blue dots). The Pearson correlation between PC2 and the *ER* signal was −0.85. The control samples that were known to be *ER*- or *ER* + had PC2 projections consistent with this interpretation (Fig. 4a, red and green dots). Additionally, a separation by *HER2* signal and an *ER*-/*HER2* + group—another expected biological signal—was visible at the border between the *ER*

subgroups (Fig. 4b, light vs. dark blue dots). A large part of PC2 therefore corresponded to the main molecular breast cancer subtypes as described by Perou[28].

PC3 projections, similarly to PC1, showed sharp variation over time (Fig. 3e) that was consistent with the behavior of control sample projections (Supplementary Fig. 2), suggesting the influence of technical signals. The two fluctuations corresponded with the use of automated sample labeling (Fig. 3e). Additionally, PC3 projections were associated with the *CLDN* signal (Fig. 4c) with a Pearson correlation of −0.5, and a Pearson correlation of 0.26 with *ER* signals, (data not visualized).

Projections of PC4 suggest an association with mainly biological factors (Figs. 3f and 4c, d). This PC showed a Pearson correlation of 0.46 with the *BRCA* signal, and −0.29 with the *CLDN* signal.

Overall, the distribution of both PC1 and PC3 fluctuated sharply over time. While projections for PC2 and PC4 also fluctuated somewhat over time, their variation was smoother than for PC1 and PC3. The three time-localized technical factors explaining PC1 and PC3 (labeling yield, automated labeling, and the combination of isolation buffer and reference RNA used) were also the ones with largest overall association with the four PCs (0.66, 0.58 and 0.53, respectively, Table 1(A)) quantified by their canonical correlation[29] (see "Methods").

The first four PCs were also associated with a milder extent with other technical factors (Supplementary Data 1) such as tumor cell percentage ($r = 0.51$) and RNA integrity number ($r =$

**Table 1 Statistical association of technical factors or biological signals before and after data adjustment.**

| Association | Unadjusted dataset | Adjusted dataset |
|---|---|---|
| (A) Canonical correlation of technical factors with PC1–4 | | |
| Automated labeling/PC1–4 | 0.66 | 0 |
| Labeling yield/PC1–4 | 0.58 | 0.27 |
| Reagent × reference RNA/PC1–4 | 0.53 | 0 |
| (B) Pearson correlation of biological signals–associations with TargetPrint (TP) readout | | |
| ER/TP-ER | 0.98 | 0.95 |
| HER2/TP-HER2 | 0.79 | 0.80 |
| PC1/TP-ER | 0.31 | 0.80 |
| PC1/TP-HER2 | 0.09 | 0.16 |

Part A lists the canonical correlation of technical factors with the top four principal components (PC1–4) before and after data adjustment. Canonical correlations are a multivariate generalization of Pearson correlation that finds the strongest possible association between two sets of variables (between linear combinations of the variables in each set). The change in statistical association between the unadjusted and adjusted data shows the disappearance or decrease of the effect of technical factors on gene expression as a result of the data adjustment. Part B gives the correlation of gene expression with ER and HER2 status that was used to assess whether biological signals were preserved after adjustment. TargetPrint (TP)-scores of the unadjusted dataset was used as a suitable substitute for the immunohistochemistry status that was not available to us at the time of analyses. The absolute Pearson correlations between ER and HER2 gene expression signals and their respective TargetPrint scores (ER/TP-ER and HER2/TP-HER2, respectively) are shown for the unadjusted and adjusted datasets. Additionally, the absolute Pearson correlations between the first principal component (PC1) and TargetPrint scores for ER and HER2 (PC1/TP-ER and PC1/TP-HER2, respectively) are shown for the unadjusted and adjusted datasets.
p-values are non-informative (see "Methods" for more information) and therefore not given. ER estrogen receptor, HER2 erb-b2 receptor tyrosine kinase 2, PC principal component, TP-ER TargetPrint for ER, TP-HER2 TargetPrint for HER2.

0.42). Importantly, these other factors were not time-localized, making it impossible to decide if they were confounded with unobserved biological factors of interest (tumor cell percentage for example could be higher in subtypes characterized by denser tumors). In general, PC1 and PC3 mainly corresponded to known technical variations and to some extent to biological signals, while PC2 and PC4 mainly corresponded to expected biological variations and seemed to be affected by smaller time-localized signals.

**Removal of technical variation (step 3).** Linking PC1 and PC3 to known technical factors helped to understand possible causes of the observed technical variation. However, adjusting the data based on these PCs could possibly lead to removing biological signals as well (see Supplementary Note 2), since PCs are not guaranteed to correspond to independent biological or technical signals only. We, therefore, adapted the *naive random RUV* method[27] to adjust for technical variation while preserving biological signals.

Our procedure fits a linear model to the expression data, with two separate terms representing unwanted variation. A first one encodes the identified time-localized factors. The second one is estimated from low-variation probes and intends to capture other sources of technical variation[27]. We shrink the estimated effects of the latter, as they could be confounded with unmodeled cancer biology. By contrast, time-localized factors are unlikely to be confounded with cancer biology and we estimate their effect with a regular regression (see "Methods" for more information). Since these technical factors clearly fluctuated over time, removing their effect was unlikely to accidentally remove biological signals. For the same reason, we chose not to explicitly adjust for other technical factors, which were not time-localized. In comparison, the regular method[27] led to less satisfactory results and poor adjustment of data (Supplementary Notes 2 and 3).

**Analysis of the adjusted dataset (step 4).** To show that the approach succeeded in removing technical variation while preserving the biological signals, we re-computed the PCs and re-analyzed their association with the known technical and biological factors. The new PC1 accounted for 5% of the total variance (Fig. 2b), while the top four PCs now explained a total variance of 15%. This relatively low percentage reflects the homogeneity of this dataset, where all tumors are from early stage breast cancer patients. After adjustment, the association of known technical factors with the top four PCs decreased substantially (Table 1 (A)). The canonical correlations[29] of the top four PCs with the technical factors of automated labeling and the combination of isolation buffer and reference RNA were reduced to zero as a direct consequence of the adjustment procedure. As labeling yield may be confounded with biological signals, we did not explicitly adjust for its effect and did not necessarily expect it to fully disappear. Data adjustment nevertheless reduced the first canonical correlation between the top four PCs and labeling yield by half to 0.27. Association with factors that we choose not to adjust for such as tumor cell percentage, and different RNA quality measurements (RIN, 28S/18S ratio, and concentration) were still present (Supplementary Data 1) and future analyses relying on our adjusted data could take these factors into account when necessary. Importantly, the projection of samples onto combinations of the top four PCs after data adjustment showed that the known biological signals were still present (Fig. 4e–h). At the same time, our adjustment removed all (isolation/labeling/scan) time-related effects, and strongly decreased all technician-related effects, without explicitly supervising any of these factors (Supplementary Data 1).

The distribution of projections onto the top four PCs computed on the adjusted dataset was also more stationary over time (Fig. 5) than the ones computed over the unadjusted dataset (Fig. 3), further indicating that the adjusted data was less affected by time-varying technical factors (also see Supplementary Fig. 3). In addition, the projections of control samples were rather constant over time in contrast to what we observed in the unadjusted data (Supplementary Fig. 2).

Correlation with ER and HER2 status was used to assess whether biological signals were indeed preserved after adjustment for unwanted technical variation. TargetPrint (TP) ($\log_{10}$ ratios of dedicated ER and HER2 probes as reported by Roepman et al.[30]) of the unadjusted dataset was used as a proxy for the immunohistochemical staining. We compared TP-readout of the dataset with ER and HER2 signals before and after data adjustment. Analyses showed high and similar correlations (Table 1(B)), indicating that their biological effect was retained in the adjusted dataset.

After adjusting for technical variation, we expected the known biological signals to explain most of the variance, which should be reflected by an increased association with the top PCs in the adjusted dataset. Consistent with this, we observed an increased Pearson correlation between PC1 and both ER- and HER2 TP-readouts (Table 1(B)). Similarly, the ER and HER2 signals (see "Methods") showed increased association with the top PCs. The Pearson correlation of ER signal and PC1 increased >4-fold to 0.85 in the adjusted data (Supplementary Fig. 3). The projection of PC1 showed a clear separation in ER signal (Fig. 4e), corresponding to ER- and ER+ subgroups that are consistent with the position of the low- and high-risk control sample projections. An ER-/HER2+ subgroup could also be distinguished in the PC1 vs. PC2 projection of the HER2 signal in the adjusted dataset (Fig. 4f), thereby explaining the limited correlation between PC1 and TP-HER2 readout (Table 1). Additionally, PC1 is correlated with the TP53 signals (0.34) and the expression signals of CLDN-associated genes (−0.36). The

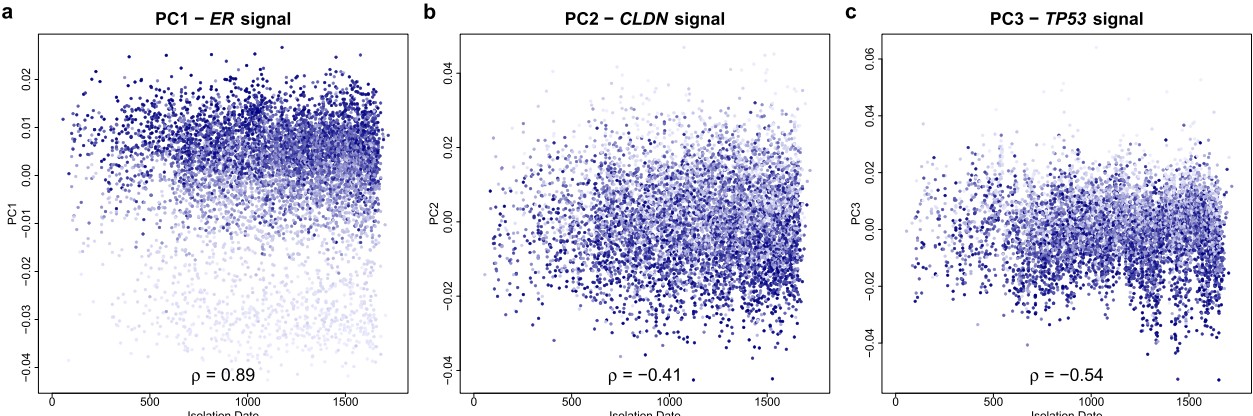

**Fig. 5 Variation over time of the top three principal components and their association with technical and biological factors in the adjusted dataset.**
With the principal component (PC) analysis we captured the largest proportion of variance across all arrays in the adjusted dataset. The top four PC projections of the unadjusted data and the adjusted data cannot be compared one to one as consequence of the adjustment itself. PC scores for the 6688 patient samples after data adjustment were plotted against time to visualize the disappearing of time-localized variation. Each dot represents one patient sample. The x-axis represents the number of days between January 1st of 2007 and the isolation date of the sample. The y-axis represents the projection of the samples onto one of the top three principal components. Color intensity of the blue dots represents the value of the plotted signal, with high color intensities corresponding to high signals (see "Methods"). The projections no longer show the fluctuations over time that were seen in the unadjusted data in Fig. 3, indicating that the technical variance causing these fluctuations was adjusted for. The biological signals remain, as indicated by a blue color intensity gradient along the PC (along the y-axis) that is especially evident for *ER* signal in the PC1 scores. *ER* estrogen receptor, *PC* principal component, *TP53* tumor protein p53.

*CLDN* signal is even stronger on PC2 (−0.41) as illustrated in Fig. 4g. Overall, the top two PCs of the adjusted data contain large effects related to well documented breast cancer subtypes (*ER +/ER−*, *HER2+/HER2−*, *CLDN* low/high). The expression of *BRCA* is associated with PC2 (0.34) and PC3 (0.35) as illustrated in Fig. 3h, while the expression of *TP53* is associated with PC3 (−0.43). Projections of the 6688 samples against the first four PCs and color-coded for *BRCA*, *CLDN*, *ER*, *HER2*, and *TP53* are presented in Supplementary Fig. 4.

In summary, variation introduced by time-localized technical factors was largely removed after adjustment while biological signals were preserved.

## Discussion

Gene expression data from studies that span multiple years naturally include variation due to technical factors, even when sources of variation are minimized by standardizing laboratory procedures and using the same batch of reagents for the duration of the study. The MINDACT study spanned several years and here we describe the MINDACT full-transcriptome microarray expression dataset and show for the first time that such a large dataset is adjusted for technical variation while preserving biological signals.

This technical variation is corrected for in the result of diagnostic tests such as MammaPrint or BluePrint[10,12] that are under diagnostic quality schemes, by using specific procedures and validations. However, its management for gene expression of the full-transcriptome is of utmost importance to avoid spurious discoveries in future analysis that may be the result of technical variation. Accordingly, the goal of this paper was to describe this technical variation, and to prepare a ready-to-use dataset for future breast cancer relevant projects by adjusting explicitly for the technical variation that could be safely modeled. Adjustment using a modified version of the *naive random RUV* method[27] implicitly removes technical variation due to known as well as unidentified sources, while preserving biological signals.

The adjusted data revealed a clear clustering of the samples in *ER+*, *ER−/HER2 +* and *ER−/HER2*-groups along the PC1, and the next principal components had a high correlation with the

expression of *TP53*, *BRCA*, and *CLDN*-associated genes. While it is impossible to guarantee that all unknown signals of interest were preserved by the adjustment, it is encouraging to see that known important signals, which were not modeled in the adjustment were preserved nonetheless. The top four principal components of the adjusted data were also much more stationary than the top PCs of the unadjusted data, suggesting that no unknown-for time-localized event had a strong influence on variance after adjustment.

The resulting dataset is representative of an early stage breast cancer population and is now available (unadjusted and adjusted) together with patient characteristics and clinical outcomes for future translational research by submitting proposals to the EORTC for access to the gene expression data as well as the clinical and patient data (https://www.eortc.org/data-sharing/), and if granted can be used for analysis and discovery science[31,32], datamining or creating new signatures of clinically relevant questions[33].

Our adjusted dataset is an ideal base for tasks such as subtype discovery[34] or inference of regulation networks[35] where the signals that should be preserved are not specified. Albeit, tasks involving a known factor of interest, e.g., creating gene signatures[36], performing differential[37] or survival analyses[38], can be addressed using either the adjusted dataset that is already controlled for technical variation, or if desirable, the unadjusted dataset, using specific procedures that deal with the unwanted variation simultaneously[13–16]. In both cases, our description of technical variation will guide future analyses to decide which additional factors should be taken into account. Not only for this dataset, but also for the analysis of future large cohort studies spanning long time periods for patient enrollment can this procedure be used as a guideline. The code that we provide is specific to the MINDACT study, but could be easily recycled on new datasets. The general pipeline described in Fig. 1 would remain identical. We recommend to always check for potential sample duplicates, and to inspect the raw data with some matrix factorization approach such as principal component analysis. In particular, we found it useful to rely on the stationarity of PCs as a proxy to their association with technical factors that only

affected gene expression over a short time window (such as automated labeling in our case, see e.g., Fig. 3). We then advise to adjust for unwanted variation, and to quantify how well the adjustment preserves known biological signals once this adjustment has been performed. Technical factors affecting expression over short time windows are unlikely to be confounded with biological signals of interest, provided that the study was homogeneous along time. Regardless of the chosen method, they can be explicitly adjusted for. By contrast, explicitly adjusting for other technical factors could also discard important, yet undescribed biological signals of interest. Instead, we relied on negative control genes to estimate unwanted variation factors, summarizing both described and potentially additional unobserved sources of technical variation, and shrank the effect of these factors to preserve biological signals. Specific points of our pipeline could be adapted, depending on the technology, existence of control samples, or the availability of technical information on the samples. The actual adjustment method is probably the most technology-dependent element of the pipeline. The technique that we used here is appropriate for microarray expression data. Extensions to metabolomic data were recently introduced[39]. Extension to count data, arising from sequencing and often modeled by Poisson or Negative Binomial (NB) distributions, is also possible. Gaussian linear models are typically used for microarray intensities, allowing the presented algorithm to rely on a least square objective, which can be derived as the maximum likelihood estimator of the linear parameter in the Gaussian model. Poisson or NB models would lead to different objectives, which could be minimized using existing toolboxes assuming the unwanted variation factor $W$ was known. In the Gaussian case, the maximum likelihood estimator of $W$ can be obtained by taking the singular value decomposition (SVD) of the expression matrix restricted to its control genes. When dealing with sequencing data, a simple option to estimate $W$ is to use SVD on the log-transformed counts for the negative control genes[40]. Alternatively, methods to compute the maximum likelihood estimate of $W$ under non-Gaussian distributions were recently introduced[41] and can be used instead.

For the gene expression dataset described in this paper, we expect the exceptional sample size and homogeneity of the data to bring an unprecedented power to differential and survival analysis, to help reveal finer subtypes and create new genomic signatures relevant to management of breast cancer.

## Methods

### Study samples

*MINDACT patients*. The MINDACT trial[3,4], sponsored by the European Organization of Research and Treatment of Cancer (EORTC) and country-associated clinical trial groups, enrolled women aged ≥18 years that were diagnosed with histologically proven operable TNM stage I–II invasive breast cancer and 0–3 positive lymph nodes. Patients were eligible to enroll in the MINDACT trial when a frozen tumor sample, containing at least 30% tumor cells, was available for gene expression analyses by microarray. For further clinical eligibility and exclusion criteria see Viale et al.[42] and Cardoso et al.[4]. Patients were enrolled in 112 participating institutions in nine European countries between February 2007 and September 2011. Tumor samples were processed in Agendia's central laboratory (Amsterdam, The Netherlands). Subsequent MammaPrint test failure rate, defined as the percentage of samples that did not finish the laboratory test procedure due to quality control failure was 1.7%.

The MINDACT trial protocol was approved by the protocol review committee of the European Organization for Research and Treatment of Cancer (EORTC) and the ethics committee at each participating site. The trial was conducted in accordance with the Declaration of Helsinki and good clinical practice guidelines. A written informed consent was obtained from all patients[4].

*Control samples*. Following diagnostic standards (FDA, ISO), multiple control samples with previously defined MammaPrint scores were used as technical and experimental controls within each batch run of samples. Data from these MammaPrint control samples were continuously monitored in the clinical diagnostic setting to ensure quality and safety[10–12] and used for quality control assessment.

This included the use for positional monitoring of array hybridization to control for potential mix-up of samples: control samples with known outcome were hybridized in a certain order and positioned on the arrays within the patient sample batches. During the time of the MINDACT trial three different MammaPrint low-risk controls and five different MammaPrint high-risk controls were used for the purpose of assessment and control of technical variations of MammaPrint procedures over time. The full-transcriptome data of the $n = 562$ control samples were only used to support visualization of the results, e.g., to show that the biological signal was still present after removal of technical variation from the dataset (see below and Fig. 4 and Supplementary Fig. 2).

### Microarray

*Microarray design*. For the MINDACT trial, a full-transcriptome microarray design was used that is based on Agilent's 4 × 44 k format (Catalog #: G2514F). This customized Agilent microarray (Agendia array 15746) was designed using annotation from genome build NCBI35. The array was customized in such a way that the MammaPrint 70 probes were added as well as additional Agendia features, counting to a total of 40,793 unique probes targeting 28,655 transcripts. The 40,793 unique probes are a combination of Agendia probes and Agilent human genome catalog probes available at the time of platform design (January 2007).

*Sample preparation and microarray hybridization*. RNA was isolated from a patient's fresh frozen tumor sample and was amplified as described previously[11]. As shown in Fig. 1a, one of the FDA/QSR-compliant technical changes involved the use of two different batches of RNA-Bee isolation reagent (Tel Test; RNA-Bee batch 1 and RNA-Bee batch 2) and one batch of Qiazol (Qiagen). A manufacturer's change in the RNA-extraction solution RNA-Bee (that was not communicated by the manufacturer) caused a temporary shift in the MammaPrint risk calculation (for more information see Cardoso et al.[4]) from May 24, 2009, to January 30, 2010, at which time the issue rectified with the use of a new reference RNA (see below). All samples of study patients ($n = 6693$) were evaluated and passed all criteria for RNA quality and RNA yield following the diagnostic MammaPrint assay-specific quality control assessment.

Cy-dyes were directly incorporated into the cRNA during in vitro transcription. A total of 750 ng of Cyanine-3 labeled patient sample RNA was co-hybridized with a Cyanine-5 labeled standard reference (dual channel hybridization) to Agendia array 15746 as described previously[11]. cRNA labeling was either performed manually or automated by use of a robot (see Fig. 1a). Arrays were subsequently washed according to the Agilent standard hybridization protocol (Agilent Oligo Microarray Kit, Agilent Technologies) and scanned with a dual laser scanner (type B, Agilent Technologies). During the study, three different reference samples were used for co-hybridization with the patient samples (see Fig. 1a). The use of MRP (Mamma Reference Pool[11]) and BP1 (Breast Pool 1), both locally assembled reference pools, was followed by the use of a commercially available UHR (Universal Human Reference; Agilent). MRP and BP1 consisted of pooled and amplified RNA of >100 primary breast tumors[11]. Samples used for MRP were isolated using RNA-Bee batch 1 and samples that were used to create BP1 consisted of samples isolated with RNA-Bee batch 2 (see Fig. 1a). BP1 was used to rectify the change in RNA-extraction solution.

**Quality control**. Agendia maintains a quality system in compliance with international regulations as FDA and the EU in vitro diagnostics directives.

*Feature extraction*. RNA expression was measured by quantifying fluorescent intensities of the scanned array TIFF-images using Feature Extraction software, version 9.5 (Agilent Technologies). Subsequent within-array normalization was performed with the lowess correction method using a linear polynomial (locally weighted linear least square regression), which was the default method for normalizing dual color Agilent microarrays. Expression values are presented as the $\log_{10}$-ratio expression values (reverted to sample/reference format). Positive and negative controls were excluded, and only positive and significant signals were used without background correction in the normalization procedure.

*Probe quality*. Only fluorescent intensities of high-quality probe measurements were used in the analysis pipeline. Probe quality was calculated by the standard Feature Extraction Software using the built-in "universal error model", and indicated by several factors including *p*-values, errors and non-uniformity indicators, and are provided in the individual sample text files.

In case the normalized red or green signals are ≤0 then the normalized $\log_{10}$-ratios were excluded from further analysis as these probes have a high inaccuracy level due to the negative value in either channel. Some probes were flagged as "Feature Non-uniformity Outliers" (FNO) in the individual sample text files and subsequently excluded from further analysis. Probes that are labeled by the software as FNO usually have a high and uneven intensity. This can be caused by a scratch, contaminant, or some manufacturer imperfection on the array, which can cause deviating ratios and is an indicator of poor quality.

*Overall full-transcriptome hybridization quality of array*. Hybridization quality was assessed, and five study samples were excluded from further analysis based on poor

quality full-transcriptome hybridizations (non-uniform hybridization). Diagnostic MammaPrint readout was not affected and passed the stringent quality assessment that was FDA/QSR controlled for accurate MammaPrint readout[11,12]. As a result, the full-transcriptome dataset contains data for 6,688 patients.

### Data preparation

*Missing value imputation.* For the 6688 patient dataset containing 40,793 probes, the number of missing values was determined. A missing value is the result of probes flagged by a poor quality as described above.

During the MINDACT study three different standard references were used (see Fig. 1a and above) in the dual channel hybridization (patient + reference sample). However, by using three different reference samples, the expression values (presented as $Log_{10}$-ratios) are not meant to be compared across the different references. Expression values were calculated based on the ratio of the sample-/ reference-intensities and can give different $log_{10}$-ratios. The number of missing values was determined and imputed for the three references (MRP, BP1, UHR) separately.

Probes missing in >20% of any of the three groups of patient arrays defined by their reference samples were omitted from further analysis ($n = 5182$) and no values were imputed, leaving 35,611 probes per patient for further analysis. Missing values were imputed for the remaining probes using a *k*-nearest neighbor approach based on other probes that behave similarly across the patient samples[43]. A separate imputation procedure was done specifically for control arrays (see Supplementary Fig. 1 for more information). Subsequently, a mean centering of the gene expression values was performed to have a mean of zero across each set of samples with the same reference, to allow for combining the three sets of samples with different references for further analyses.

*Analysis of sample correlations.* The processing of tumor samples from 6688 breast cancer patients treated in one of the 112 institutes located in nine countries involved a large number of operational steps, thereby increasing the possibility of a potential sample mix-up. Control samples with known outcome were added to each MammaPrint run and hybridized in a certain order to be able to identify potential sample mix-ups. However, possible duplication of samples cannot be identified using these positional controls. To assess whether a multiple of the reported arrays corresponded to the same sample or another sample with a highly similar profile, we studied pairwise Pearson correlations among all arrays. These correlations were computed across the probes that were not missing in any of the arrays ($n = 4703$ probes) to avoid potential introduction of artifacts from the missing value imputation procedure. To determine if a high correlation was caused by repeated processing of the same sample we reprocessed available RNA for a selected number of samples. Their MammaPrint indices were compared to the original indices (discussed in Supplementary Note 1).

### Calculation of biological signals

*Calculating gene expression signals from probe combinations.* Throughout the paper, we measure the association between principal components and expected biological signals related to the main breast cancer subtypes. The estrogen receptor (ER) signals were calculated as the average of $log_{10}$-ratios for all probes corresponding to *ESR1*, *FOXA1*, *SCUBE2*, and *PTPRT* genes. Claudin (CLDN) signals were calculated as the average expression of all probes corresponding to *CD14*, *VAV1*, *IL6*, and *VEGFC* genes. *Erb-b2* receptor tyrosine kinase 2 (HER2) signals and breast cancer gene signals were calculated as the average expression of all probes corresponding to the *HER2*, or *BRCA1*, and *BRCA2* genes, respectively.

*Proxy for* ER *and* HER2 *immunohistochemistry status.* TargetPrint (TP)-readout ($log_{10}$-ratios of dedicated *ER* and *HER2* probes[30]) of the unadjusted dataset was used as a proxy for the immunohistochemical staining. This TP-readout showed a concordance of 95% with the immunochemistry status of the unadjusted samples after binarization in Roepman et al.[30]. Viale et al.[42] showed in the first 800 MINDACT samples that the concordance between TargetPrint and immunohistochemistry was 98%, and 96% for *ER*, and *HER2*, respectively. Therefore, TP-readout is a suitable substitute for the immunohistochemistry status that was not available to us at the time of analyses.

### Statistics and reproducibility

Analyses and visualization of data was performed in R (version 3.4.4, https://www.r-project.org/[44]).

We used principal component (PC) analysis[45,46] to assess the effect of identified known technical changes and expected biological variation on mRNA expression values in the unadjusted and the adjusted dataset. PCs were computed over the patient samples across the 35,611 probes represented by their $Log_{10}$-ratios.

Pearson correlations and canonical correlations[29] were used to evaluate the association between technical or biological factors and the four PCs. Canonical correlations are a multivariate generalization of Pearson correlations that define an association between sets of variables rather than single variables. Although all associations were significant, we chose not to provide *p*-values. Indeed, the large sample size of this study makes all observed associations significant, even for very small size effects. We are interested in identifying factors, which explain a large proportion of the variance rather than testing for non-zero associations. Since we

do not perform any inference, we do not assess the normality of the factors, which is only a necessary condition to build valid inference procedures, not to use Pearson correlations as a measure of linear correlation. It is important however to assess that these associations are not affected by outliers, and that no major non-linear effect is missed. The scatter plots that we provide and discuss in the manuscript show no evidence of strong outliers. We specifically address some cases of notable non-linear associations, such as the peaks in the PC projections (section Analysis of the unadjusted expression data (step 2)), and the association between PCs and HER2 expression (section Analysis of the adjusted dataset (step 4)).

### Analyses for visualization

*Color encoding of continuous biological or technical signals.* We encoded biological signals and labeling yield using shades of blue in Figs. 3–5 (and Supplementary Figs. 3 and 4). For each encoded variable, all values were sorted, and each array was assigned a shade corresponding to its rank, with darker shades corresponding to larger values. An important consequence of using the ranks is that the proximity of colors cannot be translated into a proximity in values. This is a minor concern for our figures since we are interested in visualizing trends and clusters, not comparing pairs of arrays. We chose ranks over values because the latter was not always able to represent the full range of the variable as a single outlier value would make all the other arrays have similar color intensities.

### Data adjustment for removal of unwanted technical variation

Normalization was performed to remove technical, and not biological, variation. Data were adjusted using a procedure derived from the "naive random effect" version of remove unwanted variation (RUV) described by Jacob et al.[27]. RUV relies on a linear model of the expression data, including a factor of interest *X* and *k* unwanted variation factors stored in a matrix **W**:

$$Y = X\beta + Wa + \varepsilon,$$

where ($\alpha$, $\beta$) are the effects of the factor of interest and unwanted variation, respectively, and $\varepsilon$ is a Normal centered random variable. To estimate *W*, RUV methods rely on negative control genes, defined as genes whose expressions are not expected to be influenced by biological signals of interest. Jacob et al.[27] describe a procedure to estimate $W\alpha$ when *X* is unobserved, e.g., to adjust a dataset for future analyses with no specific factor of interest in mind. The "naive" procedure uses $X\beta = 0$. The "naive random" method additionally does a ridge regression of the gene expression matrix **Y** against these *k* factors instead of an ordinary regression to estimate $\alpha$. It refers to the fact that ridge regression provides a maximum a posteriori estimator of $\alpha$ when it is modeled as a random effect. Ridge regression limits the amount of variation captured by $\alpha$, which is crucial if there is a risk that *W* is confounded with some unobserved factor of interest *X*, especially in the naive procedure, which models $X\beta = 0$. In this study, we further regress the expression matrix against a few identified technical factors stored in a matrix **Z**: specifically, *Z* includes an interaction factor representing all combinations of reference RNA and isolation reagent (as represented in Fig. 3c), and another one representing all combinations of isolation reagent and automated labeling. We are confident that the time-localized variation caused by these factors is not confounded with biological signals. Consequently, we choose not to ridge the regression against these two time-localized factors, as we want to remove their effect entirely from the data. We rely on the estimated *W* to capture all sources of unwanted variation, which are not modeled in *Z*, including the non-time-localized technical variables described in the manuscript and other potentially unobserved technical variables. Since its effect $\alpha$ is estimated by ridge regression, the procedure only adjusts for a fraction of non-time-localized unwanted variation, leaving us the opportunity to control this fraction and assess that it does not affect known signals of interest. Formally, we jointly estimate unwanted variation by solving:

$$\min_{\alpha,\gamma} \|Y - Wa - Z\gamma\|^2 + \nu\|\alpha\|^2,$$

which has a closed form solution, then remove the estimated $W\alpha + Z\gamma$ term from *Y*.

Of note, Fig. 3 and Table 1 discuss the effect of the automated labeling factor. By contrast, our matrix **Z** represents the interaction of automated labeling with isolation reagent: as the two groups of arrays for which automated labeling was used were isolated at different time periods and with different reagents, the latter was more suited for removing the time-localized effect on gene expression.

We utilized the 1000 genes of lowest interquartile range as negative controls. We used $k = 100$ and tried several values of the $\nu$ parameter of the ridge regression by exploring a grid of values nu.coeff (0.01, 0.02, 0.05, 0.1, 0.2, 0.5, 1, 10) multiplied by the squared largest singular value of *W*. We then picked the nu.coeff, leading to the largest correlation between TargetPrint–ER[30] (see previous section) computed before adjustment and the corresponding *ER* probe after adjustment (nu.coeff = 0.2). We emphasize that the TargetPrint readouts were only used to choose the amount of variance removed, not to determine the unwanted factors *W* and *Z* in our adjustment procedure. A more detailed explanation of the choice of nu.coeff is provided in Supplementary Note 3.

The data adjustment procedure was applied to the 6688 patient samples but without the 562 control samples as the latter could bias the estimation. The data of the 562 control samples were adjusted separately using the same procedure, except that their **W** matrix was estimated on both patient and control samples, as we

wanted the adjustment of the control sample data to be closer to the one applied to the patient sample data. By construction of the $\mathbf{Z}$ matrix, both the 6688 patient sample data and 562 control sample data are mean centered by the adjustment procedure.

**Reporting summary**. Further information on research design is available in the Nature Research Reporting Summary linked to this article.

## Data availability
Both the unadjusted full-transcriptome dataset and the adjusted full-transcriptome dataset of the 6688 patients, including probe annotation, as well as a list of the technical factors per patient and TP-readout are available through the EORTC (https://www.eortc.org/data-sharing/). The MINDACT clinical trial sponsor can also provide patient characteristics and clinical outcomes. Following a successful data request procedure, the EORTC can share all or a selection of the full-transcriptome and/or clinicopathological data for future translational research.

## Code availability
A code reproducing our analysis is available along with this manuscript and is also available through the EORTC (https://www.eortc.org/data-sharing/).

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

## Acknowledgements

We would like to thank Xiaolian Yuan for supporting data management, Architha Ellappalayam for carefully reading the manuscript and the scripts, Lisette Stork for trial coordination with EORTC and Bernhard Sixt for overall oversight. Supported by grants from the European Commission Sixth Framework Program (FP6-LSHC-CT-2004-503426, to the TRANSBIG Network of Excellence), the European Breast Cancer Council–Breast Cancer Working Group (BCWG grant for the MINDACT biobank), the Breast Cancer Research foundation (for MINDACT translational Research), the Netherlands Genomics Initiative–Cancer Genomics Center (2008–2012). L.J. is funded by the Agence Nationale de la Recherche ANR-14-CE23-0003-01 (MACARON). Whole-genome analysis was provided by Agendia without cost.

## Author contributions

Authors contributed substantially to the conception (L.J., T.S., L.v.t.V., A.G.) and design (L.J., T.S., L.v.t.V., A.G.) of the work, to the acquisition (A.W., L.D., M.S., J.v.d.A., N.B.,) and analyses (L.J., I.B., A.W., J.v.d.A., A.G., D.W.), and interpretation of data (L.J., I.B., A.W., L.D., M.S., J.v.d.A., D.W., T.S., L.v.t.V., A.G.), drafting (L.J., I.B., L.v.t.V., A.G.) and revising (A.W., L.D., J.v.d.A., B.C., A.F., G.B., C.P., J.B., M.D., M.P., E.R., F.C., T.S., G.B., D.W., N.B.) the work. All authors approved the submitted version and agree to be personally accountable for their own contributions.

## Competing interests

Authors declare no competing non-financial interests, but the following competing financial interests: A.W., L.D., D.W., M.S., B.C., N.B., L.v.t.V., and A.G. are current (part-time) employees of Agendia, the company that runs the full genome microarrays and generated the gene expression data; I.B., J.v.d.A., A.F., and G.B. were employees of Agendia while engaged in the project. The authors L.J., C.P., J.B., M.D., M.P., E.M., F.C., and T.S. declare no competing financial or non-financial interests.
