## [Peer Review File · Communications Biology]

Reviewers' comments:

Reviewer #1 (Remarks to the Author):

In this paper, authors describe and assess how technical variation affect MINDACT expression dataset. To do so, first a Principal Component Analysis is conducted to describe the unwanted variation affecting the expression data of this large MINDACT dataset. Secondly, they build an adjusted version of the expression dataset for future analysis purposes which is based on an adaptation of the random naïve RUV algorithm that removes variation related to time-localized technical factors without losing biological signals. The pipeline proposed in the paper is entirely reproducible using R code.

Despite the fact that there are a wide review of methods in literature to reduce technical variation in microarray datasets, specific results obtained for MINDACT dataset are very impressive.

I would be glad to approve this document as long as authors deal with some minor methodological/statistical details (see comments in the attached files). In addition, I consider that it would be worthwhile including the reasons for which this pipeline is (or not) specific for this dataset. In case, it could be used to reduce technical noise of other datasets, please provide examples and the general pipeline.

Reviewer #2 (Remarks to the Author):

1. The data used for analysis are rather old. That is why there are microarray rather than NGS data. Please explain explicitly why only these old data were used.
2. Figures 3-5 contain much data without quantitative generalization in tables. Please think about such generalization.
3. Some parts of the text, e.g. "Data preparation and quality control", should be moved from Results to Methods section.
4. Most statements in Introduction and Discussion sections are not confirmed using References and citations. As a result, there are only 20 (twenty) items in References sections. It seems too few.
5. Please say more words about practical implications, conclusions and recommendations that may be derived from the authors' findings.

Reviewer #3 (Remarks to the Author):

The authors bring an adaptation of a pre-existing algorithm to remove technical variability while maintaining biological variability as much as possible. The analysis was performed on a collection of transcriptome data for the study of breast cancer, the MINDACT trial. Also, the authors provide this adjusted version of the datasets publicly. In general terms, the manuscript is well written, organized, and informative.

This proposal is exciting for future research in the field of the transcriptome that aims to remove technical variability, and also for studies that involve the discovery of new targets in breast cancer. However, I have some questions regarding the execution of the work and the applicability of the

methodology in future research:

- 1) I advise the authors to add a comparison of the performance of the algorithm proposed by the authors and the original version of RUV, described by Jacob et al. 2016.
- 2) The authors created a tolerance cut-off of 20% of missing probes, which is a value considered high, and could have a significant impact on the results depending on the choice of the imputation method. What led to the decision for the k-nearest neighbor? Was it tested the efficiency of other methods? If so, how was the SVD performance?
- 3) I would like to know how the authors classified the missing values (MCAR, MAR, NMAR).
- 4) What led to the choice of the mean for gene expression centralization and not the median? Was there an investigation of outliers before the decision?
- 5) I would like more explanations about the choice of the 1000 genes of lowest inter-quartile range as negative controls.
- 6) Would adaptations be necessary for the transposition of the proposed algorithm for RNA-seq data?
- 7) I believe it is essential that the authors show in the supplementary material all the PCs projections color-coded for each of the biological signals tested (ER, BRCA, HER2, and CLDN; mRNA expression level from microarray) before and after the adjustment.

REVIEWER 1

(Remarks to the Author):

In this paper, authors describe and assess how technical variation affect MINDACT expression dataset. To do so, first a Principal Component Analysis is conducted to describe the unwanted variation affecting the expression data of this large MINDACT dataset. Secondly, they build an adjusted version of the expression dataset for future analysis purposes which is based on an adaptation of the random naïve RUV algorithm that removes variation related to time-localized technical factors without losing biological signals. The pipeline proposed in the paper is entirely reproducible using R code.

Despite the fact that there are a wide review of methods in literature to reduce technical variation in microarray datasets, specific results obtained for MINDACT dataset are very impressive.

I would be glad to approve this document as long as authors deal with some minor methodological/statistical details (see comments in the attached files). In addition, I consider that it would be worthwhile including the reasons for which this pipeline is (or not) specific for this dataset. In case, it could be used to reduce technical noise of other datasets, please provide examples and the general pipeline.

Steps 1-4 in Figure 1 describe a general strategy that could be used to adjust other expression datasets. The code that we provide is specific to the MINDACT study, but could be easily recycled on new datasets. The general pipeline would remain identical, but specific points could be adapted, depending on the technology, existence of control samples, or the availability of technical information on the samples. We added a paragraph discussing the generality of our pipeline in the discussion:

The code that we provide is specific to the MINDACT study, but could be easily recycled on new datasets. The general pipeline described in Figure 1 would remain identical. We recommend to always check for potential sample duplicates, and to inspect the raw data with some matrix factorization approach, in particular to assess how known technical factors affect the data. We then advise to adjust for unwanted variation, and to quantify how well the adjustment preserves known biological signals while removing known confounders. We found it useful to rely on the stationarity of PC projections as a proxy to their association with time-localized technical factors. These factors affect gene expression (as they are associated with the first PCs) but are unlikely to be confounded with biological signals of interest, provided that the study was homogeneous along time. Regardless of the chosen method, they can be explicitly adjusted for. By contrast, explicitly adjusting for other technical factors could also discard important, yet undescribed biological signals of interest. Instead, we relied on negative control genes to estimate unwanted variation factors, summarizing both described and potentially additional unobserved sources of technical variation, and shrank the effect of

these factors to preserve biological signals. Specific points of our pipeline could be adapted, depending on the technology, existence of control samples, or the availability of technical information on the samples. The actual adjustment method is probably the most data-dependent element of the pipeline. The technique that we used here is appropriate for microarray expression data. Extensions to metabolomic data were introduced in De Livera et al., 2015. Extension to sequencing data, often modeled by Poisson or Negative Binomial (NB) distributions, is also possible. Gaussian linear models are typically used for microarray intensities, allowing the presented algorithm to rely on a least square objective, which can be derived as the maximum likelihood estimator of the linear parameter in the Gaussian model. Poisson or NB models would lead to different objectives, which could be minimized using existing toolboxes assuming the W factor was known. In the Gaussian case, the maximum likelihood estimator of W can be obtained by taking the singular value decomposition (SVD) of the expression matrix restricted to its control genes. When dealing with RNA-seq data, a simple option to estimate W would be to use SVD on the log-transformed counts for the negative control genes (a similar approach was taken in Risso et al., 2014). Alternatively, methods to compute the maximum likelihood estimate of W under non-Gaussian distributions were recently introduced (Durif et al., 2019).

Main Paper

p.5 112 institutes located in 9 countries

Final results may be affected by the moment at which data were taken. Were data previously normalize/standarized according to the institute´s time zones?

Thank you for your comment. We do not fully understand what the reviewer meant with normalized according to the institutes time zones. All institutes were located in Europe and are in the same time zone.

The RNA expression assessment, including the workflow from RNA extraction to microarray fluorescence scanning was performed in Agendia's centralized laboratory in Amsterdam. This facility works under ISO, CLIA/CAP and EU IVD CE marking and FDA oversight. The methods are fully standardized and experimental starting and stopping times are largely the same on a day-to-day basis.

If the question relates to time of day when a patient was operated and a specimen was taken for submission for MammaPrint testing at Agendia's centralized laboratories, then we do not have the data at which time of the day a specimen sample was taken and thus are unable to carry out a direct normalization for it. However, we hope/expect that if time of day has a substantial effect on gene expression, our use of negative control genes will reduce the impact of this effect.

In addition, most institutes did submit during the full Mindact trial accrual period. Any batch or time related effects are part of this study and the purpose of this study was to remove any technical effects (such as time and batch effects) to keep the biological variation.

p.6 **Moreover, Pearson correlations**

Do you rely on linear correlation? Do you check normal residuals?

We rely on Pearson correlation as a measure of linear association between variables. As discussed in the Online Methods, we choose not to provide p-values. The associations that we describe all have a large amplitude, and would lead to rejecting the null hypothesis of no association under any reasonable model. We are interested in discussing factors which explain a large proportion of the variance, as opposed to simply having a non-zero association with gene expression. While normality is important to perform inference on the measured correlations, it is not required to use Pearson correlations as a measure of linear association. It is important however to assess that these associations are not affected by outliers, and that no major non-linear effect is missed. The scatter plots that we provide and discuss in the manuscript show no evidence of strong outliers. We specifically address some cases of notable non-linear associations, such as the peaks in the PC projections (e.g., p6 in the Analysis of the unadjusted expression data (step2)), and the association between PCs and HER2 expression (p8: "An ER-/HER2+ subgroup could also be distinguished in the PC1 versus PC2 projection of the HER2 signal in the adjusted dataset (Fig. 4f), thereby explaining the limited correlation between PC1 and TP-HER2 readout").

We added a few sentences in the Online Methods when first mentioning Pearson correlations, to summarize these points:

Since we do not perform any inference, we do not assess the normality of the factors, which is only a necessary condition to build valid inference procedures, not to use Pearson correlations as a measure of linear correlation. It is important however to assess that these associations are not affected by outliers, and that no major non-linear effect is missed. The scatter plots that we provide and discuss in the manuscript show no evidence of strong outliers. We specifically address some cases of notable non-linear associations, such as the peaks in the PC projections (section Analysis of the unadjusted expression data (step2)), and the association between PCs and HER2 expression (section Analysis of the adjusted dataset (step 4)).

Please, note that you observed two peaks in the distribution corresponding to time periods, have you think on circular correlation (used to measure correlation in periodic events) instead of linear correlation. Could it be more meaningful?

The two peaks observed in Fig 3a correspond to time periods when the labeling yield was stronger and indicate the effect of this on PC1. This is not a periodic phenomenon of the kind informed by the use of circular correlation, which is appropriate to cyclic phenomena.

p.7 **Projections of PC4**

You present strong conclusions for PC4 despite that the percentage of variability that it gathers is very low, could you provide some additional evidence to confirm this fact?

Thank you for this remark. Our conclusion only regards the association of PC4 with known biological signals, and its non-association with technical factors. These observations stand regardless of the percentage of variability represented by PC4.

p.8 5%

The percentages of variability are extremely low. You should provide more evidences to reinforce your conclusions and to illustrate the improvement (e.g. RIN, concentration or other you consider). Please could you clearly detail the reasons why the percentage of variability is reduced after removing technical variation? Intuitively it should increase since artifacts are removed from biological signals.

The reported low numbers are percentages of variability explained by individual PCs, not percentages of variability explained by biological signals. In the unadjusted data, PC1 explains a rather large percentage of the total variance (Figure 2), but it is strongly associated with technical factors, and less so to biological signals. After removing the technical variation, the only variation that is left is variation due to biology. In particular, the main effect (PC1) reflects the presence of ER+ and ER- samples in the dataset (Figure 4e). The dataset is rather homogenous since all tumors are from early stage breast cancer patients, which explains why no single factor explains a large percentage of the variance (as opposed to technical factors whose effect was strong on some samples in the unadjusted data). We added a sentence in the manuscript to clarify this point (section **Analysis of the adjusted dataset (step 4)**):

“This relatively low percentage reflects the homogeneity of this dataset, where all tumors are from early stage breast cancer patients.”

(RIN, 28S/18S ratio, and concentration)

Please, include these evidences in the supplementary materia

We added these evidences as attached tables (Supplementary Data 5). We mention this addition in the manuscript next to the point highlighted by the reviewer.

p.2 lowess correction

Do you think about the effect of lowest correction method? Do you try others (e.g. median polish)? Why do you select that? How does it specifically affect on the subsequent analyses?

The lowess correction was chosen as the microarray hybridizations used dual color, for which lowess was the standard correction in the Agilent software. It is also the most widely used, and we regard it as the best, a view that has been supported by several studies (e.g.

Grant et al. 2007, Watson et al. 2007). We choose not to redo normalization in the current manuscript as we feel that any other normalization method will not influence the end result. The normalization used is specifically for the so called “within array normalization” and adjust the gene expression levels to “normal” or average levels in each array, mostly to remove dye biases.

Any batch effects that occur will not be “normalized away” by any method that only checks for normalization within each array, it will not remove any effects between arrays. In the circumstance that many samples are processed in a time period of several years/months there will be effects caused by different reagent batches or even other equipment. For that reason a correction needs to be applied to remove the technical variation as we describe in the current manuscript.

Grant GR, Manduchi E, Stoeckert CJ Jr.
Analysis and management of microarray gene expression data.
Curr Protoc Mol Biol. 2007 Jan;Chapter 19:Unit 19.6.

Watson M, Pérez-Alegre M, Baron MD, Delmas C, Dovic P, Duval M, Foulley JL, Garrido-Pavón JJ, Hulsegge I, Jaffrézic F, Jiménez-Marín A, Lavric M, Lê Cao KA, Marot G, Mouzaki D, Pool MH, Robert-Granié C, San Cristobal M, Tosser-Klopp G, Waddington D, de Koning DJ.
Analysis of a simulated microarray dataset: comparison of methods for data normalisation and detection of differential expression
Genet Sel Evol. 2007 Nov-Dec;39(6):669-83.

without background

without background

There was no particular reason not to subtract background levels from the feature signal. In general the background was low.

p.3 neighbor

Typo

Thank you for pointing this out. In the manuscript we have used American-English spelling. We will check the guidelines from the journal and will adhere to their recommendations and adjust as necessary.

p.4 Pearson

Do you have evidences about the probe distribution? Pearson correlation assumes that variables are normal distributed. Please, Could you confirm this fact?

The normality assumption is necessary to produce valid inference on the association between two variables. Here we simply use it to scan for potential mix-ups (samples which may have been profiled twice by mistake), without drawing any inference. We clarified this aspect in a new paragraph in the Online Methods (please see above our answer to your other comment on normality p.6 of the main manuscript).

p.5 $Y = X\beta + W\alpha + \epsilon$

Please, could you explain the similarities/differences between RUV and a classical random effects model? Do you use the term ‘naïve procedure’, but I think it is far to be naïve, it is a combination of different statistical procedures, do you think about solve estimation problem from the random effect model?

RUV methods differ from classical random effects models in that they have a step equivalent to factor analysis, which is neither fixed nor random in the usual of jargon of linear models. “Naïve” may be inappropriate, but we use it for consistency with Jacob *et al.* 2016 where the method was first described. It refers to the assumption that $X\beta=0$. The main difference with usual linear models is that the neither X term, representing signals of interest, nor the W term representing technical factors, are assumed to be observed. In the general case, X and W could not be distinguished, and maximum likelihood estimation on this model would recover matrix decomposition methods such as PCA. RUV relies on negative control genes, i.e., genes whose expression is expected to be mostly unaffected by X, to estimate W (by matrix decomposition on Y restricted to these genes). W is then plugged back into the model, and α is estimated by ridge regression of W against Y (this is where we use the naive assumption that $X\beta=0$). The ridge is used because X (which is unobserved) could be confounded with W, in which case a regular regression would remove all signal associated with X. Here, we added a third term $Z\gamma$ representing known technical factors that we want to adjust for. Concretely, Z contains an interaction factor representing all combinations of reference RNA and isolation reagent (as represented in Figure 3c), and another one representing all combinations of isolation reagent and automated labeling. By contrast, W is expected to capture other, unspecified sources of unwanted variation. We observe that our adjustment does reduce the effect of technical factors that were not in Z, as discussed in the **Analysis of the adjusted dataset (step 4)** section of our manuscript.

Supp file 1

p.1 **To assess whether any of of the reported arrays may correspond to the same sample,**

It is a vague statement. Correlation analysis explores relationship among variables. Please, be more precise.

Thank you for your comment. We performed a pairwise correlation using the gene expression values as variables. Gene expression values were determined by the fluorescent probe intensities of 4,703 probes. The sentence was changed to “To assess whether any of the reported arrays may correspond to the same sample, we studied pairwise Pearson correlations among all pairs of arrays, each array being represented by a vector with one entry per measured probe.”

p.7 **over the time.**

Going one step further, do you think if it could be related to the circadian clock in some way? Many biological rhythms display rhythmic (periodic) patterns over time

Thank you for your comment. Although an interesting suggestion, we do not think that the pattern we see is related to the circadian clock. The control samples that were used are from the same RNA stock (isolated only one time), and processed multiple times in a row. Therefor we don’t expect to see biological rhythms or patterns in control samples, only technical factors.

p.8 that PC1 in

Why do you only mention PC1?

I think that conclusion for unadjusted and adjusted data should be discussed together (only one subsection). Conclusions are not clear for the reader.

Thank you for your suggestion. In the adjusted data all PCs are likely to present biological variation in the samples. We only mention PC1 for the adjusted data as it separates the MammaPrint high and low risk control samples nicely indicating prognosis factors such as ER and Her2. The other PCs represent other (unknown) biological processes.

We created two subsections to better distinguish between the two datasets. The panels on the left represent the unadjusted data and the show the first 4 principal components. After the technical variation is removed and dataset is adjusted, the principal component analysis is performed again and results in new PC1 to PC4. We would prefer to retain the current structure with two subsections, to avoid confusing readers between PCs obtained over the two versions of the data.

REVIEWER #2

(Remarks to the Author):

1. The data used for analysis are rather old. That is why there are microarray rather than NGS data. Please explain explicitly why only these old data were used.

The Mindact study was run between 2007 and 2012, but the expression data has not been made available until now. Moreover, the clinical follow-up data collection is still ongoing and therefore over time the dataset is becoming richer for data analysis. Such datasets do not exist elsewhere, either with microarray or NGS data. Our study is the first to describe this dataset, as well as its adjustment. The size and homogeneity of this study are unprecedented, and we believe that it provides a relevant illustration of issues that can arise when managing large scale gene expression studies.

2. Figures 3-5 contain much data without quantitative generalization in tables. Please think about such generalization.

We discuss a few specific quantities in the text: the association between principal components and technical factors or biological signals. We selected these quantities because we believed that they were the most relevant for understanding how the dataset was structured, before and after adjustment. All associations are reported in the tables described in Supplementary Data 5. Please note that we are not performing any inference nor are we considering any generalization: both the figures and extracted quantities are descriptive statistics.

3. Some parts of the text, e.g. "Data preparation and quality control", should be moved from Results to Methods section.

Our article describes a pipeline to adjust for technical variation in a large study, and discusses the application of this pipeline to Mindact. The actual methods used at each step of the pipeline are indeed described in the Methods section, but we also feel that presenting the outcome of each step in the Results section is important. Accordingly, we removed the sentences

“Quality and consistency of all experimental and data processing analytic steps were monitored by a quality system that is compliant with international regulations as FDA and the EU in vitro diagnostics directives. Data quality was further assessed as described in the Online Methods.”

from “Data preparation and quality control”, but would like to leave the rest in Results.

4. Most statements in Introduction and Discussion sections are not confirmed using References and citations. As a result, there are only 20 (twenty) items in References sections. It seems too few.

Thank you for this comment. We added references to support our statements in Introduction and Discussion (there are now 41 in the main manuscript, and 46 after merging with the Online Methods).

5. Please say more words about practical implications, conclusions and recommendations that may be derived from the authors' findings.

We added to the discussion a paragraph discussing the generality of our procedure, its most important elements and how it could be extended to other types of data:

The code that we provide is specific to the MINDACT study, but could be easily recycled on new datasets. The general pipeline described in Figure 1 would remain identical. We recommend to always check for potential sample duplicates, and to inspect the raw data with some matrix factorization approach, in particular to assess how known technical factors affect the data. We then advise to adjust for unwanted variation, and to quantify how well the adjustment preserves known biological signals while removing known confounders. We found it useful to rely on the stationarity of PC projections as a proxy to their association with time-localized technical factors. These factors affect gene expression (as they are associated with the first PCs) but are unlikely to be confounded with biological signals of interest, provided that the study was homogeneous along time. Regardless of the chosen method, they can be explicitly adjusted for. By contrast, explicitly adjusting for other technical factors could also discard important, yet undescribed biological signals of interest. Instead, we relied on negative control genes to estimate unwanted variation factors, summarizing both described and potentially additional unobserved sources of technical variation, and shrank the effect of these factors to preserve biological signals. Specific points of our pipeline could be adapted, depending on the technology, existence of control samples, or the availability of technical information on the

samples. The actual adjustment method is probably the most data-dependent element of the pipeline. The technique that we used here is appropriate for microarray expression data. Extensions to metabolomic data were introduced in De Livera et al., 2015. Extension to sequencing data, often modeled by Poisson or Negative Binomial (NB) distributions, is also possible. Gaussian linear models are typically used for microarray intensities, allowing the presented algorithm to rely on a least square objective, which can be derived as the maximum likelihood estimator of the linear parameter in the Gaussian model. Poisson or NB models would lead to different objectives, which could be minimized using existing toolboxes assuming the W factor was known. In the Gaussian case, the maximum likelihood estimator of W can be obtained by taking the singular value decomposition (SVD) of the expression matrix restricted to its control genes. When dealing with RNA-seq data, a simple option to estimate W would be to use SVD on the log-transformed counts for the negative control genes (a similar approach was taken in Risso et al., 2014). Alternatively, methods to compute the maximum likelihood estimate of W under non-Gaussian distributions were recently introduced (Durif et al., 2019).

REVIEWER #3

(Remarks to the Author):

The authors bring an adaptation of a pre-existing algorithm to remove technical variability while maintaining biological variability as much as possible. The analysis was performed on a collection of transcriptome data for the study of breast cancer, the MINDACT trial. Also, the authors provide this adjusted version of the datasets publicly. In general terms, the manuscript is well written, organized, and informative.

This proposal is exciting for future research in the field of the transcriptome that aims to remove technical variability, and also for studies that involve the discovery of new targets in breast cancer.

However, I have some questions regarding the execution of the work and the applicability of the methodology in future research:

1) I advise the authors to add a comparison of the performance of the algorithm proposed by the authors and the original version of RUV, described by Jacob et al. 2016.

We provide such a comparison in Supplementary Notes 6 and 7.

2) The authors created a tolerance cut-off of 20% of missing probes, which is a value considered high, and could have a significant impact on the results depending on the choice of the imputation method. What led to the decision for the k-nearest neighbor? Was it tested the efficiency of other methods? If so, how was the SVD performance?

K-nearest neighbor was previously reported to outperform SVD (Missing value estimation methods for DNA microarrays, Troyanskaya *et al.*, Bioinformatics 2001), and we are not aware of cases where the SVD had a clear advantage on k-nearest neighbors. The 20% threshold already led to discarding 13% of the probes. Since we were able to provide a

version of the data with this threshold removing technical effects and retaining biological signals, we chose not to discard more probes, to make the dataset as versatile as possible. Researchers interested in a subset of the probes could run the process again with a lower threshold.

3) I would like to know how the authors classified the missing values (MCAR, MAR, NMAR).

We would classify the missing values as MNAR. mRNAs with lower expression are more likely to lead to missing values than highly expressed mRNAs.

4) What led to the choice of the mean for gene expression centralization and not the median? Was there an investigation of outliers before the decision?

In almost all microarray gene expression studies, attention focusses on mean log fold changes. The use of medians in this context is extremely rare, and hard to implement. Further, the feature extraction software only retained log ratios between -4 and 4 (these were the default parameters), which excludes the possibility of an outlier causing a large deviation of the mean.

5) I would like more explanations about the choice of the 1000 genes of lowest inter-quartile range as negative controls.

We investigated both using the lowest inter-quartile range and a set of 658 control genes identified by Agendia. The rationale for using low-variation genes was that they were less likely to be affected by biological signals. We obtained similar results as with the control genes, and therefore chose to retain the more generic inter-quartile approach. Importantly, we directly assessed the quality of the adjusted data, as opposed to the quality of the negative controls. The latter is a more difficult, and to our opinion an unnecessary objective.

6) Would adaptations be necessary for the transposition of the proposed algorithm for RNA-seq data?

Sequencing count data are often modeled by Poisson or Negative Binomial (NB) distributions, whereas Gaussian linear models are typically used for microarray intensities. The current algorithm relies on a least square objective, which can be derived as the maximum likelihood estimator of the linear parameter in the Gaussian model. Poisson or NB models would lead to different objectives, which could be minimized using existing toolboxes assuming the W factor was known. In the Gaussian case, the maximum likelihood estimator of W can be obtained by taking the singular value decomposition (SVD) of the expression matrix restricted to its control genes. When dealing with RNA-seq data, a simple option to estimate W would be to use SVD on the log-transformed counts for the negative control genes (a similar approach was taken in "Normalization of RNA-seq data using factor analysis of control genes or samples", Risso *et al.*, Nature Biotechnology 2014). Alternatively, methods to compute the maximum likelihood estimate of W under non-

Gaussian distributions exist, see *e.g.* “Probabilistic Count Matrix Factorization for Single Cell Expression Data Analysis”, Durif *et al.*, Bioinformatics 2019.

We added a few words on this point in our Discussion.

7) I believe it is essential that the authors show in the supplementary material all the PCs projections color-coded for each of the biological signals tested (ER, BRCA, HER2, and CLDN; mRNA expression level from microarray) before and after the adjustment.

We added these figures to the supplementary material (Supplementary Figure 9), and mention this addition in the description of Figure 4 (which Supplementary Figure 9 extends), as well as the end of the **Analysis of the adjusted dataset (step 4)** section.

REVIEWERS' COMMENTS:

Reviewer #1 (Remarks to the Author):

In this paper, authors describe and assess how technical variation affect MINDACT expression dataset. To do so, first a Principal Component Analysis is conducted to describe the unwanted variation affecting the expression data of this large MINDACT dataset. Secondly, they build an adjusted version of the expression dataset for future analysis purposes which is based on an adaptation of the random naïve RUV algorithm that removes variation related to time-localized technical factors without losing biological signals. The pipeline proposed in the paper is entirely reproducible using R code.

Despite the fact that there are a wide review of methods in literature to reduce technical variation in microarray datasets, specific results obtained for MINDACT dataset are very impressive.

I am glad to approve this document. Authors have accurately dealt with the pending methodological/statistical details. One the one hand, they have included new paragraphs, both in the main and supplementary files, explaining most issues required (pipeline, Pearson correlation and so on). One the other hand, they have thoroughly answered all of my questions in the rebuttal letter. Moreover, the potential of this work allows to pose new questions as those related to the gene expression's time of the day.

Reviewer #2 (Remarks to the Author):

The following new paragraph on pp. 10-11, starting with

"The code that we provide is specific to the MINDACT study, but could be easily recycled on new datasets."

and ending with:

"Alternatively, methods to compute the maximum likelihood estimate of W under non-Gaussian distributions were recently introduced"

seems to long and not so much comprehensive.

Please clarify, what "confounders", "time-localized technical factors", "PC projections" etc. are meant. The practical examples or figures may be useful.

Reviewer #3 (Remarks to the Author):

The authors successfully answer the questions, making the suggested changes, or presenting a strong argument.

REVIEWERS' COMMENTS:

Reviewer #1 (Remarks to the Author):

In this paper, authors describe and assess how technical variation affect MINDACT expression dataset. To do so, first a Principal Component Analysis is conducted to describe the unwanted variation affecting the expression data of this large MINDACT dataset. Secondly, they build an adjusted version of the expression dataset for future analysis purposes which is based on an adaptation of the random naïve RUV algorithm that removes variation related to time-localized technical factors without losing biological signals. The pipeline proposed in the paper is entirely reproducible using R code.

Despite the fact that there are a wide review of methods in literature to reduce technical variation in microarray datasets, specific results obtained for MINDACT dataset are very impressive.

I am glad to approve this document. Authors have accurately dealt with the pending methodological/statistical details. One the one hand, they have included new paragraphs, both in the main and supplementary files, explaining most issues required (pipeline, Pearson correlation and so on). One the other hand, they have thoroughly answered all of my questions in the rebuttal letter. Moreover, the potential of this work allows to pose new questions as those related to the gene expression's time of the day.

We thank the reviewer for carefully reviewing our manuscript and helpful suggestions.

Reviewer #2 (Remarks to the Author):

The following new paragraph on pp. 10-11, starting with

"The code that we provide is specific to the MINDACT study, but could be easily recycled on new datasets."

and ending with:

"Alternatively, methods to compute the maximum likelihood estimate of W under non-Gaussian distributions were recently introduced"

seems to long and not so much comprehensive.

Please clarify, what "confounders", "time-localized technical factors", "PC projections" etc. are meant. The practical examples or figures may be useful.

We thank the reviewer for pointing out this unclear section. We provide a new version where

previously undefined terms have been removed, and where we re-organized some of the ideas to make the message more clear. The updated sections are highlighted in green.

Reviewer #3 (Remarks to the Author):

The authors successfully answer the questions, making the suggested changes, or presenting a strong argument.

We thank the reviewer for carefully reviewing our manuscript and helpful suggestions.